# Lips Are Lying: Spotting the Temporal Inconsistency between Audio and Visual in Lip-Syncing DeepFakes

**Weifeng Liu**[1,*], **Tianyi She**[1,*], **Jiawei Liu**[1], **Boheng Li**[2], **Dongyu Yao**[3], **Ziyou Liang**[1], **Run Wang**[1,†]

[1] Key Laboratory of Aerospace Information Security and Trusted Computing,
Ministry of Education, School of Cyber Science and Engineering, Wuhan University, China

[2] Nanyang Technological University

[3] Carnegie Mellon University

{weifengliu, tianyishe, jiaweiliu, wangrun}@whu.edu.cn
boheng001@e.ntu.edu.sg
raindy@cmu.edu

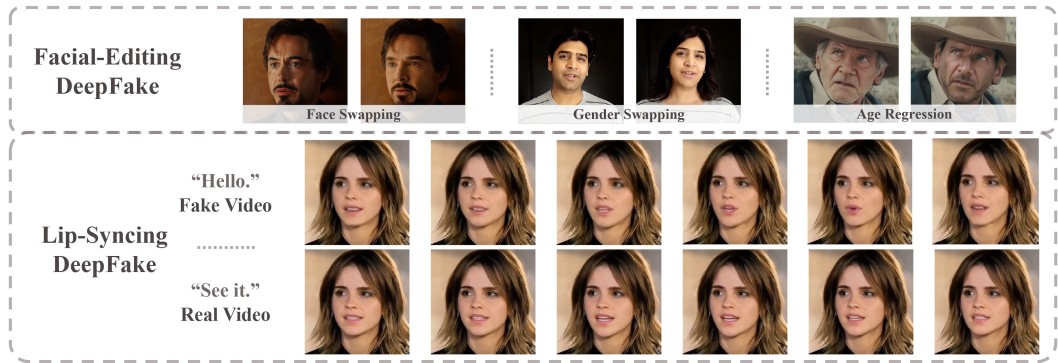

Figure 1: A visualization comparison between common deepfakes and our studied lip-syncing deepfakes (LipSync). The former exhibits a substantial forgery area and identity manipulation, such as face or gender swapping, whereas the latter, relies on the synchronization of the minor lip region and given audio, without any alterations to the subject's identity. As illustrated in the comparison above, discerning the authenticity of an image sequence becomes arduous in the absence of labels.

## Abstract

In recent years, DeepFake technology has achieved unprecedented success in high-quality video synthesis, but these methods also pose potential and severe security threats to humanity. DeepFake can be bifurcated into entertainment applications like face swapping and illicit uses such as lip-syncing fraud. However, lip-forgery videos, which neither change identity nor have discernible visual artifacts, present a formidable challenge to existing DeepFake detection methods. Our preliminary experiments have shown that the effectiveness of the existing methods often drastically decrease or even fail when tackling lip-syncing videos. In this paper, for the first time, we propose a novel approach dedicated to lip-forgery identification that exploits the inconsistency between lip movements and audio signals. We also mimic human natural cognition by capturing subtle biological links between lips and head regions to boost accuracy. To better illustrate the effectiveness and advances of our proposed method, we create a high-quality LipSync dataset, AVLips,

---

*Equal contribution.

†Corresponding author.

by employing the state-of-the-art lip generators. We hope this high-quality and diverse dataset could be well served the further research on this challenging and interesting field. Experimental results show that our approach gives an average accuracy of more than 95.3% in spotting lip-syncing videos, significantly outperforming the baselines. Extensive experiments demonstrate the capability to tackle deepfakes and the robustness in surviving diverse input transformations. Our method achieves an accuracy of up to 90.2% in real-world scenarios (*e.g.,* WeChat video call) and shows its powerful capabilities in real scenario deployment. To facilitate the progress of this research community, we release all resources at https://github.com/AaronComo/LipFD.

# 1   Introduction

DeepFake refers to an AI-based technology for synthesizing fake media data [1]. The recent advancements in generative models, particularly the emergence of several GAN architectures [2, 3, 4] and the diffusion probabilistic models [5], have enhanced the realism and quality of forged videos that can easily deceive humans. The prevalence of DeepFake poses potential security risks, *e.g.,* political elections and identity verification, sparking public concerns [6].

DeepFake can be bifurcated into entertainment applications and illicit uses [7]. As illustrated in Fig. 1, the popular DeepFake aims to bring fun to users by swapping faces to synthesize new content, such as gender swapping and age regression. Unfortunately, the severe DeepFake is utilized for illicit crimes, including manipulating political propaganda and fabricating pornographic content. The case is particularly alarming in LipSync fraud, where the audio drives the mouth movements on reconstructed video frames. These DeepFakes are generally exploited by malicious actors in real-world scenarios, such as the widely disseminated fabricated videos of Barack Obama saying things he never said on YouTube [8], posing significant security threats. The escalating issue of real-time forgery necessitates an effective detector to identify videos generated through LipSync.

Unlike popular DeepFake which manipulates facial attributes or replaces the entire face, LipSync does not tamper with identity and possesses subtle visual artifacts. More seriously, attackers can adaptively erase these visual artifacts through blurring. Since LipSync follows visual modification driven by audio modality, detecting LipSync forgeries naturally involves spotting the inconsistencies between lips and audio. Whereas the correlation between them is closely tied to individual talking styles, intensifying the challenges in developing a universal model to represent this correlation.

Existing studies on DeepFake detection can be classified into unimodal-based and multi-modal-based methods, where the former relies on visual discrepancies arising from identity tampering to detect [9, 10, 11, 12]. However, unimodal detectors become less reliable when the forged videos are perturbed for targeted removal of LipSync artifacts. In recent years, several multi-modal-based methods have emerged [13, 14, 15], including audio-visual fusion and audio-visual inconsistency. Fusion strategies may confound the learning of singular modality features and the performance post-fusion is not necessarily enhanced [16]. [17] suggested training detectors to learn the inconsistencies between video frames and audio. However, as the arms race between DeepFake creation and detection intensifies, these inconsistencies are gradually reduced, making coarse-grained audio-visual alignment strategies less effective against advanced LipSync methods.

Lip movements are discrete, while the audio spectrum is continuous, resulting in inherent inconsistencies in LipSync videos. As illustrated in Fig. 2, we observe a temporal correlation between the energy variations in spectrum and lip movements. To the best of our knowledge, existing works naively align single-frame images with long-range audio clips, thus neglecting the temporal inconsistencies of audio-visual features [18, 19]. Our experimental evidence also indicates a marked decline in the efficacy of existing methods when confronting LipSync forged videos. Moreover, [20] demonstrated the mouth region's significance in facial appearance, surpassing even the eyes, due to its biological connections with other head regions. Humans naturally leverage the cues of local regions and head postures to discern facial semantics. While DeepFake technology has made strides in replicating overall facial dynamics, it often falls short of accurately simulating these subtle yet crucial biological interactions. Hence, we choose to exploit the biologically intrinsic correlation between lip movements and head postures as auxiliary information to detect deepfakes. This approach not only mimics the natural cognitive processes of humans but also capitalizes on the existing limitations of deepfakes.

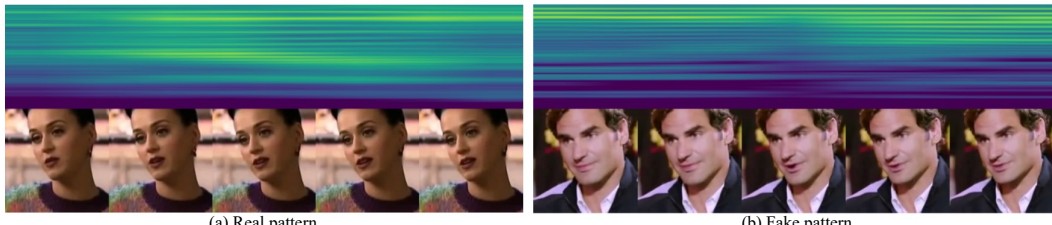

|  |  |
|:---:|:---:|
| (a) Real pattern | (b) Fake pattern |

Figure 2: **(a)** shows the correlation between lip movements and corresponding spectrogram in genuine pattern. When the woman starts talking, the middle and high frequencies in the spectrum are lighted. Over time, the energy gradually fades and shifts from middle to lower frequencies. **(b)** the first two frames show a highlighted high-frequency spectrum, contradicting the man not speaking. In the third frame, an unexpected lip opening appears at the darkest part of the spectrum. The mouth cannot change so drastically within a single frame, and this lip shape contradicts the spectrum information.

In this paper, for the first time, we propose **LipFD**, a pioneering method that leverages the inconsistencies in audio-visual features for the **Lip**-syncing **F**orgery **D**etection. Specifically, our approach captures irregular lip movements that contradict the audio signal aligned with it in the temporal sequence of audio-visual features. We also devise a novel framework that dynamically adjusts the attention of LipFD to regions with different clipping ratios.

To evaluate the effectiveness and generalization of our approach in detecting lip-syncing deepfakes, we utilize the state-of-the-art LipSync methods to generate massive high-quality lip forgery video dataset based on Lip Reading Sentences 3 (LRS3) [21], Face Forensics++ (FF++) [22], Deepfake Detection Challenge Dataset (DFDC) [1]. Experimental results show that our approach outperforms prior works by a notable margin, with an accuracy up to 96.93% for four types of lip forgery videos. Rigorous ablations of our design choices and comparisons with other detection methods demonstrate the superiority of our approach. Our main contributions can be summarized as follows:

- We propose the first-of-its-kind approach dedicated to lip-syncing forgery detection that is often overlooked by existing studies. This method addresses the significant and growing threat of lip-syncing frauds, like those encountered in WeChat video calls.

- In this work, we unveil a key insight that exploits the discrepancies between lip movements and audio signals for fine-grained forgery detection. Our approach introduces a dual-headed model architecture to enhance detection capabilities.

- We construct the first large scale audio-visual LipSync dataset with up to 340,000 samples, and conducted comprehensive experiments on it alongside other DeepFake datasets. Our method demonstrated high efficacy and robustness, achieving around 95% average accuracy in LipSync detection, and up to 90.18% in real-world scenarios.

## 2 Related Work

**Lip-syncing Generation.** Lip-syncing facial manipulation, which forges a speaker's lip movements to match a given audio, is among the most threatening DeepFake applications due to its subtlety and difficulty to detect, typically falsifying the speaker's conveyed information. [23] disentangled the content and speaker information in the audio signal, allowing attackers to generate a forged video using just a single image and an audio segment. Still, it is weak in representing bilabial and fricative sounds due to the omission of short phoneme representations. [24] introduced a well-trained discriminator and a temporal consistency checker to address the loss of short-duration phoneme, enhancing the authenticity of generated videos. However, it exhibits weak temporal coordination in the lip movement of talking heads. [25] further focuses on the content of lip movements, making the forgeries challenging for both human eyes and machines to recognize.

**DeepFake Detection.** The existing DeepFake detectors employ single-modal-based or multi-modal-based approaches to detect subtle differences between real and fake samples. Earlier single-modal detectors aspired to employ neural networks to automatically extract discriminative information [26, 27], but they failed to detect unseen samples due to overfitting. To address this issue, some

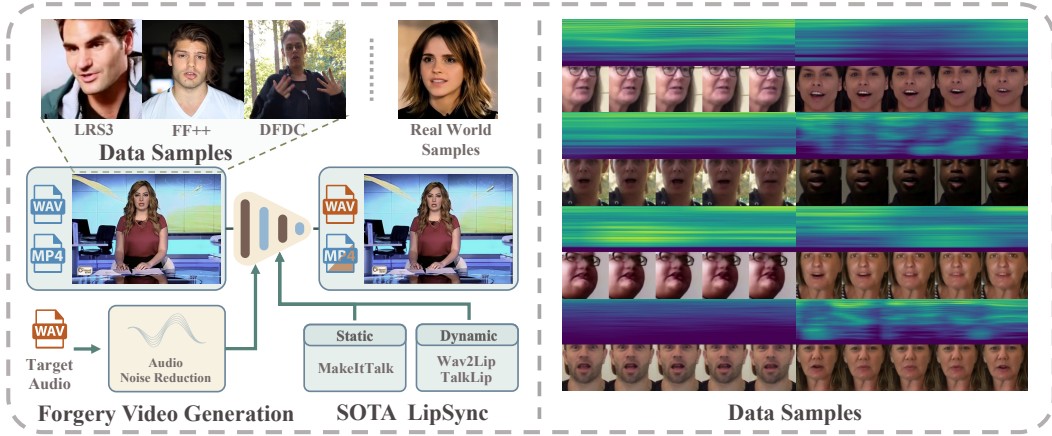

Figure 3: **AVLips dataset construction.** Utilizing static and dynamic methods, we generated high-quality videos with realistic lip movements. The diverse dataset includes various real-world scenarios. Perturbations were applied for robust model training.

studies shift focus to frequency domain features [28, 29] or subtle forgery artifacts in more generalized datasets [30, 31]. Another line is to guide the network to focus on discriminative locations, such as automatically guiding the detector's positional attention through a double-stream network [32], or manually cropping the lip region to extract artifacts formed by the inconsistent lip movements [33]. Although these works have achieved considerable performance on afore datasets, they are not sensitive when faced with advanced lip-syncing generators due to the absence of synchronized audio features. In the multi-modal-based detectors, noticing that the coordination of audio-visual modalities is an inherently challenging issue in any SOTA generator, [34] quantifies the disparity between audio and visual as the criterion for classification, but focusing too much on the background information in the video led to failure. In this context, [35] intentionally extracts talking head movements and establishes a correlation with audio for discrimination. These methods performed well in addressing audio-visual forgery, but are susceptible to the influence of noise or compression.

## 3   LipSync Forgery Dataset

To the best of our knowledge, the majority of public DeepFake datasets consist solely of videos or images, with no specialized one specifically dedicated to LipSync detection available. To fill this gap, we construct a high-quality **A**udio-**V**isual **Lip**-syncing Dataset, **AVLips**, which contains up to 340,000 audio-visual samples generated by several SOTA LipSync methods. The workflow is demonstrated in Fig. 3.

**High quality.** We employed a combination of static 'MakeItTalk' [23] and dynamic 'Wav2Lip' [24], 'TalkLip' [25], 'SadTalker' [36] generation methods to simulate realistic lip movements. These methods are widely recognized as high-quality work, capable of generating high-resolution videos while ensuring accurate lip movements. We applied a noise reduction algorithm to all audio samples before synthesis to reduce irrelevant background noise, ensuring the models can focus on speech content.

**Diversity.** Our dataset encompasses a wide range of scenarios, covering not only well-known public datasets but also real-world data. Our aim is for this collection to act as a catalyst for advancing real-time forgery detection. To better simulate the nuances of real-world conditions, we have employed six perturbation techniques — saturation, contrast, compression, Gaussian noise, Gaussian blur, and pixelation — at various degrees, thus ensuring the dataset's realism and practical relevance.

## 4   Method

In reality, the movements of a speaker's lip and head are closely intertwined with the spoken content, forming a natural and coherent unity. These physical movements naturally align with the timing and

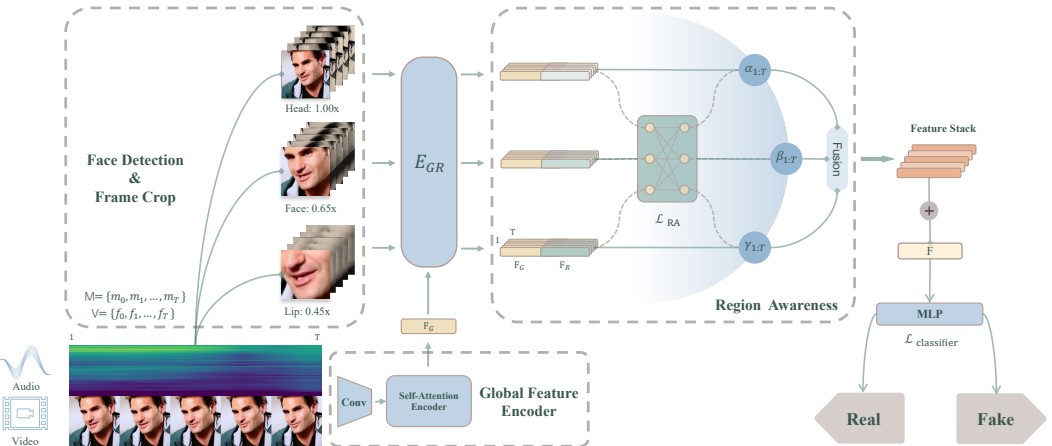

Figure 4: **Overview of LipFD framework.** Blue components represent our main modules in LipFD. The input image was generated by pre-processing, which consists of $T$ frames in the target video and their audio spectrogram. (a) The aim of Global Feature Encoder, a self-attention model, is to extract long-term information between video frames and audio, finding unreasonable correspondences between lip movements and audio. (b) $E_{GR}$ encodes three series of crops, focusing on different parts for each region, and concatenates them with global feature $F_G$. (c) The Region Awareness module assigns corresponding weights to the features based on their importance. (d) All features are fused together into a unified representation $F$ based on their respective weights for final inference.

context of the speech. However, LipSync method, which solely relies on audio signals to generate lip movements frame by frame, only focuses on the precise alignment between lip shapes and speech at any given moment. It overlooks the broader temporal context and the overall coherence of lip and head movements during speech. Consequently, the generated outputs often exhibit inherent inconsistencies regarding temporal synchronization. These inconsistencies serve as valuable clues and insights for our detection efforts, highlighting the disparity between natural lip movements and artificially generated ones. Fig. 2 vividly exhibit the temporal features among those two classes.

Extracting temporal inconsistencies between audio and video presents notable challenges due to the utilization of features from multiple modalities. To tackle this, we developed a dual-headed detection architecture presented in Fig. 4. (1) The Global Feature encoder is dedicated to encoding temporal features, capturing the overarching correlation between audio and lip movements. (2) The Global-Region encoder aims to detect subtle visual forgery traces within regions of varying scales and integrate them with global features. (3) Moreover, we introduced an innovative Region Awareness module that dynamically adjusts the model's attention across different scales. We will demonstrate in Sec. 6.1 that this module stands as a cornerstone, harnessing features from regions of diverse sizes, thus empowering our model to effectively capture both the prominent changes in DeepFake and the subtle adjustments in LipSync.

### 4.1 Global Feature Encoding

Based on the findings mentioned before, we need to extract features in the temporal domain. Inspired by the translation task in natural language processing, where transformers detect long-distance vocabulary correlations, we regard the inherent correlation between lip movements and spectral information as analogous to the relationship between 'vocabulary' in a 'sentence' sequence. To capture and encode this correlation, we employ a transformer model.

To effectively carry out its task, the encoder necessitates extraordinary representational capacity, which can be attained through exposure to a vast number of images [37]. This capacity enables the encoder to accurately allocate attention to the relevant regions of interest. To satisfy this requirement, we choose a variant of vision transformer ViT:L/14 [38], pre-trained on CLIP [39]. In our experiments, we use the final layer of CLIP: ViT-L/14's visual encoder for image embedding.

**Formulation.** We denote the convolutional layer as $Conv$, which convolves images down to $224 \times 224$. We first crop source image $I$ into 3 series as $\{c_h^N, c_f^N, c_l^N\}_i$, $i \in \{0, ..., T-1\}$, where $N$

equals to batch size $T$ notes the window size, $c_l$ is the lip region subject to modifications by LipSync, $c_f$ represents face area focused on by DeepFake, and $c_h$ encompasses the overall zone containing head posture and background information. The three-tiered cropping strategy emulate the human visual focus on key facial areas, spotlighting the lip and overall facial structure. Image $I$ will be embedded into $F_G$ as global feature:

$$F_G = ViT(Conv(I)) \tag{1}$$

$$\{c_h^N, c_f^N, c_l^N\}_i = Crop(I, \{1.0, 0.65, 0.45\}), \ i \in \{0, 1, 2\} \tag{2}$$

The encoder is constrained by $\mathcal{L}_{RA}$ that is to be further described in the following section.

## 4.2 Region Awareness

LipSync tends to concentrate on the lower half of the face. Relying solely on coarse-grained global features is insufficient for representation. Hence, we use local features to better capture forgery traces.

**Formulation.** For each crops $c \in \{c_h^N, c_f^N, c_l^N\}_i$, region feature is defined as $F_R = E_{GR}(c, F_G, \theta_{GR})$. We hope this component can focus on the most informative parts of different cropped regions, i.e. lip for $c_l$ and head pose for $c_f, c_h$. Since lip forgery is often slightly manipulated only on the mouth, the unsupervised model may fail to learn proper representation. We further introduce a region awareness module that applies a modified fully connected layer followed by a sigmoid function, which takes both sub-regions within crops as well as pertinence between region features and their relevant global context into consideration, thus granting different weights to them. The weight is formulated as:

$$\omega_{c_j^i} = RA([F_G | \{F_R\}_j^i]; \theta_{RA}), \ c_j \in \{c_h, c_f, c_l\} \tag{3}$$

where $c_j^i$ denotes the i-th feature in $c_j$ and $\theta_{RA}$ is the parameters of region awareness module $RA(\cdot)$. The final feature $F$ is obtained by concatenating the global feature $F_G$ with three series of region features $F_R$, which represent the relation between temporal features and region visual features:

$$F = \frac{1}{T} \cdot \frac{\sum_{i,j}(\omega_{c_j^i} \cdot [F_G | \{F_R\}_j^i])}{\sum_{i,j} \omega_{c_j^i}} \tag{4}$$

**Region Awareness Loss.** We noticed that, regardless of the high-level patterns learned by the model, it is the lower part of the face that matters most [40, 41]. Other extracted information should be served as auxiliary. Hence, we designed $\mathcal{L}_{RA}$, encouraging the region awareness module to focus more on areas that are more frequently modified. Mathematically, the loss is defined as:

$$\mathcal{L}_{RA} = \sum_{j=1}^{N} \sum_{i=1}^{T} \frac{k}{\exp([\omega_j^i]_{max} - [\omega_j^i]_h)} \tag{5}$$

where $\omega_{max}^i$ is the max weight in feature stacks, $\omega_h^i$ is the none-cropped region. $k$ is a hyper-parameter used to adjust the steepness of the loss. With $\mathcal{L}_{RA}$, we hope the model can focus on areas with a higher probability of being modified, such as the face and lips.

## 4.3 Lip Forgery Detection

According to Eq. 4, the crop with the highest weight exerts dominance over the feature $F$, indicating that it encapsulates crucial discriminative information for the final detection.

**Classification.** We implement a multi-layer perceptron [42] as our classifier optimized with a Binary Cross-Entropy loss:

$$\mathcal{L}_{cls} = -(y \log(F) + (1 - y) \log(1 - F)) \tag{6}$$

where $y$ means the final predicted label. Finally, the objective is given by:

$$\min_{\theta_{RA}, \theta_{cls}, \theta_{GR}} \omega \cdot \mathcal{L}_{RA}(\theta_{GR}, \theta_{RA}) + \mathcal{L}_{cls}(\theta_{cls}) \tag{7}$$

Table 1: **Cross-datasets validation.** Results on AVLips, FF++, and DFDC are reported, including *acc*, *ap*, *fpr*, and *fnr*. The best result is highlighted in bold, while the second-ranking one is underscored. Throughout the entire experiment, the threshold for the AP metric was set to 0.5.

| Method | AVLips | | | | FF++ | | | | DFDC | | | |
|---|---|---|---|---|---|---|---|---|---|---|---|---|
| | ACC | AP | FPR | FNR | ACC | AP | FPR | FNR | ACC | AP | FPR | FNR |
| CViT | 65.54 | 56.68 | 0.07 | 0.61 | 62.86 | 54.17 | 0.24 | 0.50 | 70.99 | 58.06 | 0.06 | 0.50 |
| DoubleStream | 75.52 | 67.72 | 0.13 | 0.36 | 91.02 | 87.64 | **0.03** | 0.14 | 77.39 | 69.28 | 0.21 | 0.24 |
| UniversalFakeDetect | 50.03 | 50.02 | 0.99 | **0.01** | 50.43 | 50.16 | 0.99 | *0.01* | 49.86 | 49.94 | 0.98 | **0.01** |
| SelfBlendedImages | 49.99 | 52.13 | 0.07 | 0.51 | 64.59 | 57.93 | 0.17 | 0.53 | 48.47 | 49.06 | 0.15 | 0.50 |
| RealForensics | 91.78 | 90.14 | **0.02** | 0.14 | 93.57 | 91.32 | **0.03** | 0.10 | 92.54 | **91.62** | **0.00** | 0.15 |
| LipForensics | 86.13 | 81.56 | 0.18 | 0.10 | 94.03 | **93.25** | 0.04 | 0.08 | 90.75 | 87.32 | 0.08 | 0.11 |
| **LipFD (Ours)** | **95.27** | **93.08** | 0.04 | 0.04 | **95.10** | 76.98 | 0.06 | 0.05 | **94.53** | 78.61 | 0.08 | 0.04 |

Table 2: **Cross-manipulation generalisation.** Evaluation scores when videos are exposed to various unseen forgery algorithms.

| Method | ACC | FPR | FNR | AUC |
|---|---|---|---|---|
| Wav2Lip (Dynamic) | 95.27 | 0.04 | 0.04 | 95.27 |
| MakeItTalk (Static) | **96.93** | **0.02** | **0.03** | **96.89** |
| TalkLip (Dynamic) | 79.33 | 0.34 | 0.04 | 80.36 |

Table 3: **Overall ablation results regarding core modules.** We evaluated our model's performance after removing components listed in the left column.

| Component | ACC | AP | FPR | FNR | AUC |
|---|---|---|---|---|---|
| Global Encoder | 95.07 | 91.81 | 0.02 | 0.07 | 95.09 |
| Global-Region Encoder | 72.52 | 64.38 | **0.01** | 0.53 | 72.50 |
| Region Awareness | 76.45 | 72.65 | 0.38 | 0.09 | 76.32 |
| Full model | **95.27** | **93.08** | 0.04 | **0.04** | **95.27** |

# 5 Experiment

## 5.1 Setup

**Datasets.** We trained our model on Wav2Lip-modified LRS3, a subset of our proposed AVLips. We evaluated our method performance on the following datasets: (1) FF++ [22], which contains 2,000 samples. (2) DFDC [1], which has 500 samples. (3) AVLips, our proposed dataset, which includes more than 20,000 samples. Since the baselines we compared against were primarily trained on the FF++ or DFDC datasets, to ensure fairness in the evaluation, we regenerated synthetic data for the first two datasets during the testing phase. This approach aims to maintain consistency and provide a level playing field for a fair comparison of the results.

**Metrics.** Following existing works [43, 37, 44], we adopt four popular metrics to get a comprehensive performance evaluation of LipFD. Specifically, we report ACC (accuracy), AP (average precision), FPR (false positive rate), and FNR (false negative rate). We use the AUC (area under the curve) as a metric to evaluate the performance in tackling various perturbation attacks.

**Baselines.** We take the SOTA methods in general DeepFake detection and lip-based detection as baselines. (1) For image-based DeepFake detections, UniversalDetect [37], DoubleStream [32] and SelfBlendedImages [31] are selected. (2) For video-based DeepFake detections, CViT [26] and RealForensics [35] are considered. (3) For the lip-based detection method, we employ the latest LipForensics [33]. Detailed information can be found at our Appendix. B.

## 5.2 Effectiveness Evaluation

In evaluating the performance of LipFD in detecting LipSync manipulation and the generation across different forgery techniques as well as obtaining a comprehensive performance evaluation, we use four different metrics to report the detection rate and false alarm rate.

Table 1 shows the performance of LipFD and prior works. We take the most advanced general DeepFake detection methods SelfBlendedImages and UniversalFakeDetect, along with representative DoubleStream and CViT video stream detection models as the baseline for DeepFake detection. We also compared our method with the SOTA lip-based method, namely LipForensics, which guides facial judgment through lip pre-reading. In addition, we have also compared the SOTA multi-modal detection method, namely RealForensics. Experimental results demonstrate that LipFD outperforms all competitors to a significant extent with a high detection rate and low false alarm rate in detecting

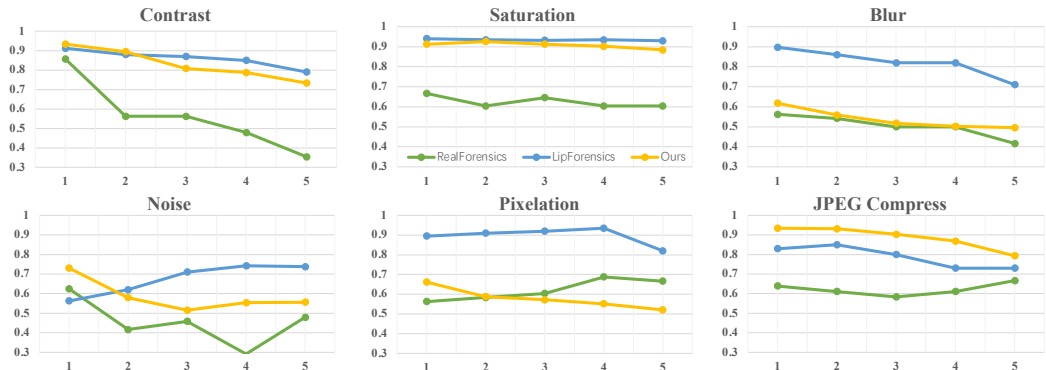

Figure 5: **Robustness against various unseen corruptions.** Average AUC scores across five intensity levels for various corruptions. For detailed analysis, please refer to the appendix.

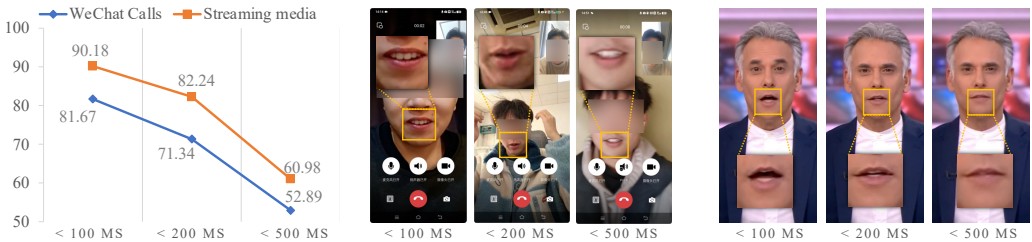

Figure 6: **Performance in real scenarios.** The x-axis represents network delay time, where a higher delay indicates a degradation in image transmission quality and clarity. Consequently, this degradation adversely impacts the audio-video synchronization in WeChat video calls.

the three DeepFake datasets. Also, we find that LipFD attains a commendable precision, as evident from the AP metric. Furthermore, we observe some discernible patterns from Table 1.

First, advanced manipulations are hard to detect by general methods such as UniversalFakeDetect and SelfBlendedImages, indicating that single-frame-based detectors cannot capture dynamic forgeries. In addition, compared to the SOTA RealForensics method, our ACC exceeded it by 3.49%, 1.7%, and 2.73%, respectively. Similar improvements are reflected in the AP as well. This illustrates that concentrating on lip-syncing allows for the extraction of more potential discriminative features than solely observing lip-based movement.

We observe some bad cases from Table 1. For example, the AP score is 16.27% lower than LipForensics on the FF++ dataset. On the DFDC dataset, our method has an AP lower than LipForensics and RealForensics by 13.01% and 12.14%, respectively. As an explanation, these methods primarily aimed at detecting large-scale manipulations of faces in these types of forgeries. In contrast, LipFD focuses on subtle changes in lip inconsistency. Despite a subtle decrease in balance, LipFD still achieves optimal performance in terms of accuracy.

### 5.3 Generalizability to Unseen Forgery

A qualified detector should recognize fake videos generated by unseen methods. We analyze our model's generalizability using the protocol from [43, 37, 44].

Table 2 shows the results of LipFD on various types of methods. Surprisingly, our detector performs even better on data generated by the MakeItTalk method than on the training data itself. This is because MakeItTalk generates dynamic videos by transforming single static images, which inherently lack the coherence of real lip movements. When we use temporal audio-visual information for joint discrimination, it becomes easier to distinguish between real and fake videos.

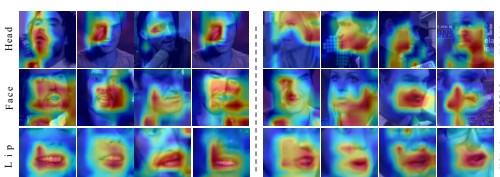
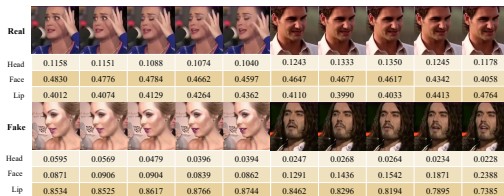

Figure 7: **Content sensitivity.** Left is real, right is fake. Visualization of the gradients from the last layer of the Global-Region encoder, which reflects the regions LipFD relies on.

Figure 8: **The weights assigned by Region Awareness.** A higher value indicates that the corresponding region of the image has a more significant impact on the final feature vector.

## 5.4 Robustness Evaluation

Robustness analysis aims to evaluate the capability of detectors to withstand common perturbation attacks, as corruptive manipulations on videos are prevalent in the wild, especially in the case of forged videos. Following the setup of RealForensics [35], we train the model on AVLips without data augmentation and then discuss the robustness of the detectors by testing with unseen samples exposed to a set of perturbations. We investigate the performance of the detectors under six types of perturbation at five varying intensities. We use AUC score as evaluation metric, and the experimental results are presented in Fig. 5. Evidently, our method outperforms the latest and the best DeepFake detectors RealForensics on most perturbation types.

Our approach effectively against saturation and contrast perturbations which performing linear transformations in the HLS space. For compression, LipFD exhibits less corruption under varying levels of quality. Gaussian blur is applied with a fixed kernel size, adjusting the standard deviation for intensity. Both blurring and pixelation significantly degrade detector performance by disrupting high-frequency information.

## 5.5 Performance in Real Scenarios

With the advancement of LipSync, certain forgery techniques have been employed for fraudulent purposes. To assess the practicality of our model in real-world scenarios, we conducted experiments across diverse network environments. Our model achieved up to 90.18% accuracy in a network with latency below 100ms which is the common situation of daily life [45, 46, 47]. Results are shown in Fig. 6. For more details, please refer to our Appendix. D.

## 6 Ablation Studies

### 6.1 Core Modules

Table 3 shows the overall situation of the experiment. Three significant components, Global feature encoder ($E_G$), Global-Region encoder ($E_{GR}$), and Region Awareness module, are ablated from the network separately, and their respective impact on the overall framework was reflected through changes in accuracy metric.

**Global-Region encoder.** Global-Region encoder takes cropped images and a vector encoded by the Global feature encoder as input, merging them into latent codes representing the correlation between regional parts and temporal sequence. The encoder $E_{GR}$ plays a crucial role in the model. As shown in Table 3, there is a significant drop in performance when $E_{GR}$ is ablated. In Fig. 7 we visualized the gradients from the last layer of it using Grad-Cam. In the third line with tag 'lip', the area near the lip has the deepest red color representing the highest gradient. The model focuses precisely on the shape of the entire lips. Meanwhile, in the above two lines, the encoder directs its attention to other features, specifically positional information, primarily on the bottom of the heads regardless of real or fake samples, for the reason that LipSync methods predominantly manipulate the bottom parts.

**Region awareness.**    The module assign different weights to the feature stack based on their contributions to discriminator. These features are then fused into the final feature, with higher weights

Table 4: **Performance under different ViT structures.** We selected six popular vision transformers as Global Feature Encoder and tested the final performance of our model.

| ViTs | ACC | AP | FPR | FNR |
|---|---|---|---|---|
| CLIP:ViT/L14 | **95.27** | **93.08** | 0.04 | **0.04** |
| CLIP:ViT/B16 | 95.00 | 92.05 | 0.03 | 0.07 |
| ImageNet:ViT/L16 | 93.28 | 91.13 | 0.09 | **0.04** |
| ImageNet:ViT/B16 | 93.27 | 91.13 | 0.09 | **0.04** |
| ImageNet:Swin-B | 94.66 | 92.53 | 0.06 | **0.04** |
| ImageNet:Swin-S | 94.59 | 90.71 | **0.02** | 0.09 |

indicating greater influence. Representative weights are shown in Fig. 8, normalized as follows:

$$\omega_i = \frac{w_i}{\omega_h + \omega_f + \omega_l}, \ \omega_i \in \{\omega_h, \omega_f, \omega_l\} \tag{8}$$

For the majority of forged video clips, the crops tagged as 'lip' are assigned significantly higher weights than other regions, indicating that these parts contain the most crucial contextual information for discrimination. On the contrary, our module leverages more information in larger-scale images ('face' and 'head') to form the latent code.

## 6.2 Selection of the Vision Transformers

In this section, we give a comprehensive view of the selection of the ViTs regarding different pretrained datasets and structures. Results are demonstrated in Table 4.

With the same architecture, the parameter count has a relatively small impact on final performance. Larger pretrained datasets and more challenging pretraining tasks lead to superior model performance and more balanced recognition capabilities (reflected in small differences in False Positive Rate and False Negative Rate). This aligns with our statement in the paper: 'To effectively carry out its task (capture temporal features), the encoder necessitates extraordinary representational capacity, which can be attained through exposure to a vast number of images'

Under the same pretrained dataset, more advanced model architectures typically lead to better final performances. For example, Swin Transformer achieves better results than vanilla ViTs. This is possibly because the window-based approach employed by Swin Transformer is more suitable for capturing long-term dependencies in video data, assisting in better identification of temporal features.

## 7 Conclusion

In this paper, we proposed LipFD, the first approach by exploiting temporal inconsistencies between audio and visual to detect lip forgery videos. LipFD demonstrates its efficacy in achieving high detection rates while exhibiting fabulous generalization to unseen data and robustness against various perturbations. We contribute AVLips, a high-quality audio-visual dataset for LipSync detection to the community, aiming to foster advancements in the domain of forged video detection. We hope our study encourages future research on lip-syncing DeepFake detection.

## 8 Acknowledgement

This research was supported in part by the National Key Research and Development Program of China under No.2021YFB3100700, the National Natural Science Foundation of China (NSFC) under Grants No. 62202340, 62372334, the CCF-NSFOCUS 'Kunpeng' Research Fund under No. CCF-NSFOCUS 2023005, the Open Foundation of Henan Key Laboratory of Cyberspace Situation Awareness under No. HNTS2022004, Wuhan Knowledge Innovation Program under No. 2022010801020127, the Fundamental Research Funds for the Central Universities under No. 2042023kf0121.

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

# A AVLips Dataset

In this section, we will provide a detailed description of the features. We have introduced the first audio-visual dataset specifically designed for LipSync detection. The goal of this dataset is to solve the issue of many existing DeepFake datasets lacking audio, while also providing foundational support for the field of lip forgery detection.

## A.1 Features

**Dynamic expansion.** The raw dataset consists of video files in MP4 format and audio files in WAV format. The videos are manipulated using state-of-the-art LipSync generation methods to forge lip movements. The original dataset can be dynamically expanded up to 50 times its initial size using the provided preprocessing code. The expanded samples are represented in the format shown in Fig. 9. The randomness algorithm ensures that each expansion generates unique data, ensuring data diversity and providing a convenient data processing approach for temporal detection methods.

**Real-world simulation.** Since our ultimate goal is to achieve real-time detection in the real world, we employed seven perturbation methods listed in Table 1, introducing various levels of perturbations to the images, to generate a substantial amount of robust training data. In addition, we have collected real-world samples from the internet, encompassing different scenes and varying levels of clarity to further enhance the diversity and realism of the dataset.

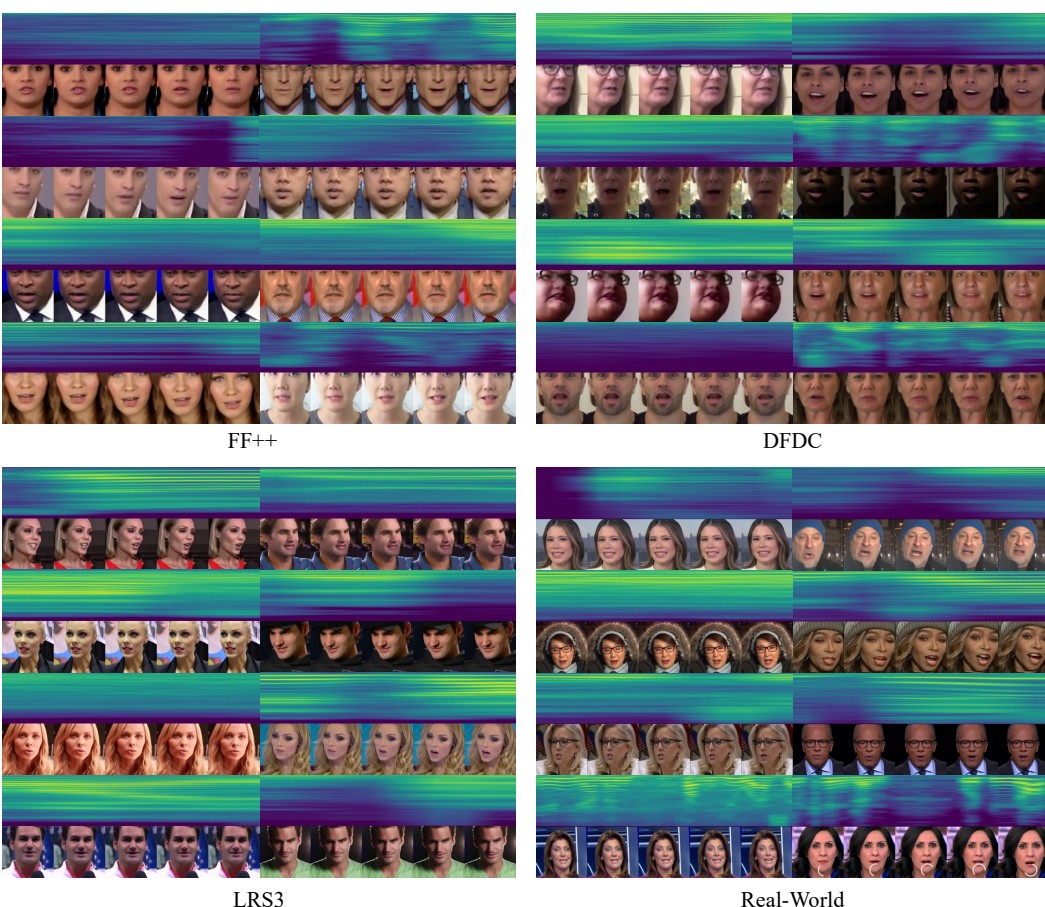

Figure 9: **Expended data samples.** Each sample consists of $T$ frames of video images and their corresponding audio spectra, serving as a temporal representation of the audio-visual context.

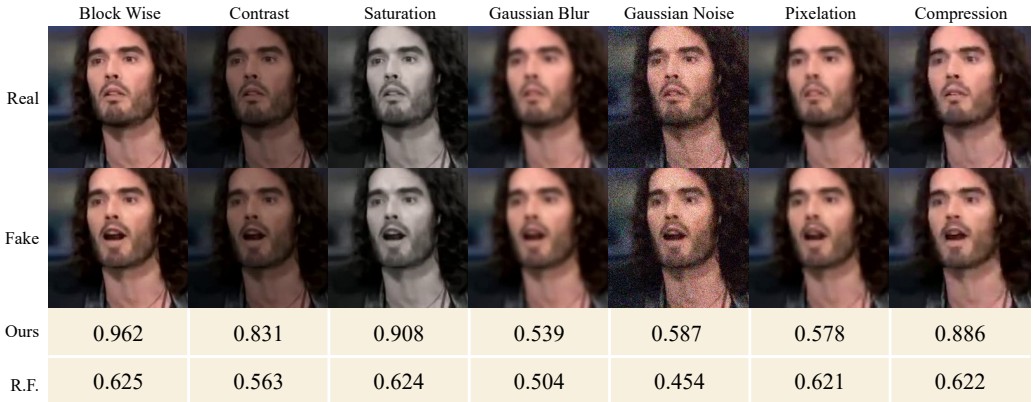

| | Block Wise | Contrast | Saturation | Gaussian Blur | Gaussian Noise | Pixelation | Compression |
|------|-----------|----------|------------|---------------|----------------|------------|-------------|
| Ours | 0.962 | 0.831 | 0.908 | 0.539 | 0.587 | 0.578 | 0.886 |
| R.F. | 0.625 | 0.563 | 0.624 | 0.504 | 0.454 | 0.621 | 0.622 |

Figure 10: **Perturbed samples and average results.** Real / Fake videos are corrupted using common perturbation methods at intensity level 3, followed by the extraction of video frames to obtain samples. Average AUC is the evaluation metric, indicating better robustness of detectors with higher values. R.F. stands for RealForensics detection.

## B Experiment Setup

To ensure a fair comparison, we have collected the only known method that employs lipreading from a high dimensional semantic viewpoint for enhancing DeepFake detection (LipForensics [33]), alongside top-performing recent DeepFake detectors. In addition, to fairly compare the performance of methods integrating both video and audio signals, we have collected the latest method that addresses video artifacts and audio-visual inconsistencies (RealForensics [35]), which currently stands as the most effective audio-assisted detection method based on our research. The two methods above are both pretrained on FaceForensics++. We evaluated them simultaneously on LipSync and DeepFake detection tasks using well-known datasets FF++, DFDC, and our proposed AVLips.

We ensured fair training settings across all baselines, with a batch size set to 32, the optimizer and the learning rate strictly adhering to the original paper's settings (Adam, 1e-3∼1e-6). Specifically: 1) Since LipForensics did not provide training scripts, we directly used the pre-trained checkpoints and performed inference under the same experimental settings. For other models, we fine-tuned pretrained weights on AVLips to meet the same baseline; 2) SelfBlendedImages, as an image-based DeepFake detector, was trained with random frames extracted from videos, following the original paper's settings; 3) UnivefsalFakeDetection, also an image DeepFake detection model, we preprocessed video data into single-frame images for training. Featuring a ResNet core with a Vision Transformer for temporal encoder, our model has approximate 310M parameters pretrained model size, which is comparable to the baselines listed in Table 1.

## C Robustness Evaluation

Due to the vulnerability of videos in the wild to varying degrees of corruptions, it is imperative for detectors not only to possess exceptional generalization capabilities but also to withstand common perturbations to accurately identifying fabricated videos. In this context, we investigate the performance of detectors under seven types of perturbations, each at five different intensity severity.

**Setup.** In our work, conducted without data augmentation, we train on the LRS3, FF++, and DFDC datasets and then expose test samples to previously unseen perturbations to examine the robustness of our detector. These perturbations encompass block-wise distortion, variations in contrast and saturation, blurring, Gaussian noise, pixelation, and video compression. As illustrated in Table 1, the block-wise changes the number of blocks, with a higher count indicating more severe distortion. Contrast and saturation are manipulated by altering the percentage of chrominance and luminance in video frames, where lower values correspond to greater corruptions. The blurring process entails adjustments to the size of the Gaussian kernel, and Gaussian white noise alters the variance of noise

Table 5: **Robustness experiment parameters.** Each perturbation method employs five unique sets of hyperparameter values, modifying them solely during the video preprocessing phase.

| Type | Hyperparameter | Severity | | | | |
|---|---|---|---|---|---|---|
| | | 1 | 2 | 3 | 4 | 5 |
| Block-wise | Block number | 16 | 32 | 48 | 64 | 80 |
| Color Contrast | Pixel value | 0.85 | 0.725 | 0.6 | 0.475 | 0.35 |
| Color Saturation | YCbCr channel | 0.4 | 0.3 | 0.2 | 0.1 | 0.0 |
| Gaussian Blur | Gaussian kernel size | 7 | 9 | 13 | 17 | 21 |
| Gaussian Noise | Noise variance | 0.001 | 0.002 | 0.005 | 0.01 | 0.05 |
| Pixelation | Pixelation Level | 2 | 3 | 4 | 5 | 6 |
| Compression | Constant Rate Factor | 30 | 32 | 35 | 38 | 40 |

Table 6: **Evaluation of Real-world scenarios.** Detection accuracy under various network delays and languages. CH stands for Chinese, and EN is in short for English.

| Latency | 100ms | | 200ms | 500ms |
|---|---|---|---|---|
| Language | CH | EN | EN | EN |
| WeChat video calls | 72.53 | 81.67 | 71.34 | 52.89 |
| Streaming media | 74.41 | 90.18 | 82.24 | 60.98 |

values. Video compression employs a constant rate factor to measure the ratio of video quality to size, with higher values denoting increased compression ratio.

**Results Analysis.** In the absence of any perturbations, the state-of-the-art detector, RealForensics [35], exhibits performance that is second only to our method. However, Figure 1 shows a significant decline in the performance of RealForensics across the majority of perturbation types, whereas our method remains efficacious under most corruptions. Perturbations involving contrast and saturation engage the percentage of chrominance and luminance in the HLS space, where our detector maintains high AUC values, suggesting an effective retention of detection capabilities in diminished visual quality. However, the detector encounters a moderate decline in performance under conditions of blurring, Gaussian noise, and pixelation, though it still surpasses the RealForensics. This indicates the noise and reduced resolution impact the detector's ability to accurately discern authenticity, potentially due to the refuction of high-frequency information. As for video compression, our approach exhibits remarkable resilience, achieving an average AUC of 0.886, which underscores the capacity of our detector to maintain high performance even when videos are subject to substantial compression, such as in real-world digital communications.

## D  Real-world Scenario

To better demonstrate the effectiveness of our proposed method in tackling the real threat of LipSync which is prevalent in the video call or financial frauds, we design and carry out extensive experiments to illustrate its practicability.

The quality of video calls and streaming clarity in the real world heavily relies on the quality of the network connection. Numerous applications employ algorithms such as ABR (Adaptive Bitrate) to dynamically adapt the audio and video bitrate and clarity, taking into account the user's network conditions. However, this adaptive process may inadvertently introduce visual blurring and noise to the video. Additionally, network latency results in network jitter and packet loss and disrupts the synchronization between audio and video [48, 49], posing significant obstacles for accurate LipSync detection in real-world settings.

### D.1  Setup

In order to simulate real-world network environments, we conducted experiments using an Android device with root access. Using Android Traffic Control, we imposed strict network conditions on the

devices, recording 1-minute English and 1-minute Chinese videos under network latencies of 100ms, 200ms, and 500ms, all at a resolution of $2340 \times 1080$. Each video was segmented into 5-second clips and dynamically expanded tenfold during preprocessing.

**D.2 Performance**

Table 6 displays the accuracy of our model in two different real-world scenarios, aligning with the information presented in the line graph in the main text.

It is evident that the accuracy of our model in WeChat video calls is generally lower compared to streaming videos. This discrepancy can be attributed to our model's strong reliance on the inconsistency between audio and video. As mentioned in our earlier analysis, as the latency increases, the audio gradually lags behind the video, creating a natural time difference that impedes our model's performance. Conversely, in streaming videos, network latency primarily affects video bitrate and clarity, resulting in blurriness and noise reduction, while not significantly altering the synchronization between audio and video. Consequently, our model exhibits better overall performance in the task of streaming videos as opposed to video calls.

Apart from network condition, language also associates with performance. Videos with Chinese language under normal network condition result in much lower accuracy. Chinese and English have distinct pronunciation characteristics. The syllable and phoneme structures in Chinese differ from English. Moreover, Chinese has a flatter intonation pattern, while English exhibits more pitch variation and prosodic contours. These phonetic and prosodic differences can impact linguistic patterns, making the correspondence between lip movements and audio more complex in Chinese videos, thereby reducing the accuracy of LipSync detection, as LipFD relies on the consistency between lip movements and audio spectrum.

# E  Discussion and Future Work

Our method not only achieved high accuracy on the LRS3, FF++, and DFDC datasets but also demonstrated its effectiveness in real-world evaluations under normal network conditions. However, the performance of our model decreases considerably when faced with Non-English-speaking videos as mentioned in Sec. D.2. So, it is crucial to incorporate multilingual training data to improve its accuracy and robustness in handling diverse linguistic contexts.

With the significant advancements in instant communication and large vision models (LVMs), generative models have made progress in achieving real-time cross-language forgery [50, 51]. The necessity of deploying a real-time LipSync detection system has come into the spotlight. Two research directions hold great promise:

- **Multilingual LipSync Detection.** Expanding LipSync detection to include multiple languages is an important area for future exploration. Investigating the challenges and differences in lip movements across various languages can contribute to developing more robust and accurate multilingual LipSync detection models.
- **Real-Time LipSync Detection.** Enhancing LipSync detection algorithms to operate in real-time scenarios is another significant research direction. Real-time detection is crucial for applications such as live streaming and video conferencing. Developing efficient and accurate algorithms that can process and analyze audio and video in real-time will be essential for these applications.

