# OpenReview forum: "Lips Are Lying: Spotting the Temporal Inconsistency between Audio and Visual in Lip-Syncing DeepFakes"
_NeurIPS.cc/2024/Conference — NeurIPS 2024 poster_

### Official Review · Reviewer_Ze8K · 2024-06-20

**Soundness:** 2
**Presentation:** 1
**Contribution:** 4
**Rating:** 6
**Confidence:** 4

**Summary:**

This paper tackle deepfake detector problem with audio-visual data focusing on lipsync fake which generally a higher quality fake data. For that, this paper propose a dataset and a method. The dataset (AVLips) is formed using available datasets and 3 methods for the lipsync methods. The method (LipFD) extracts global and local features. For global features, transformer is utilized to get the context from all regions. For local regions, the video is cropped to different face areas: face + background, face, lips areas; and extract feature from each area. Weighting is used to select which cropped areas are more important.

**Strengths:**

1. Contributions (dataset and method) in addressing lipsync based deepfake look sufficient.

2. Fine-grained features are considered.

3. Analysis in real scenarios is interesting.

**Weaknesses:**

1. Ablation removing one or two out of the 3 branches for local feature extraction is missing. Figure 8 is just showing the important weight extracted from the overall framework doesn't show exactly how much the performance drop if the branches are not included.

2. Details are not clear (see Questions).

**Questions:**

1. Line 267: is there statistics how often the latency below 100ms happen?

2. Does this work utilize audio data or not? Figure 4 bottom-left and Line 83-84 indicate audio data is used but I can't find anything related to audio input in the equations.

3. Eq. (5): Where is j index in the equation? And what is none-cropped region?

4. Line 185: \textit{notice} from where?

5. Figure 8: why lip is not as important in real data? I assume synchronized lip is also sign of real.

**Limitations:**

I can't find the limitations section. In the checklist, there should be limitation section however the reference to the section is broken.

---

> ### Author Rebuttal · Authors · 2024-08-07
>
> Dear Reviewer Ze8K, we are genuinely grateful for your valuable feedback. We sincerely hope our clarifications below can address your concerns.
>
> ---
>
> **W1:** Ablation removing one or two out of the 3 branches for local feature extraction is missing.
>
> **R1:** Thank you for your suggestion. Following your comment, we have conducted ablation experiments for the three branch structures.
>
> |Removed|ACC|AP|FPR|FNR|
> |-|-|-|-|-|
> |None|95.27|93.08|0.04|0.04|
> |Head|90.63|84.15|0.00|0.18|
> |Face|86.32|84.12|0.23|0.04|
> |Lip|89.22|87.61|0.19|0.03|
> |Face + Lip|64.15|62.11|0.69|0.03|
> |Head + Lip|82.97|82.39|0.33|0.01|
> |Head + Face|74.05|65.78|0.03|0.47|
>
> - **Head:** Contains features present in real videos. Retaining only this branch increases the FPR as the model classifies more samples as real. This region covers a larger area and contains more coarse-grained features, leading to lower model accuracy.
> - **Face:** Contains features from both real and forged videos, with more real features. It provides rich facial (e.g., expressions) and lip movement information. The model's accuracy is lower when retaining only this branch compared to retaining two branches, showing the need for additional information from other regions.
> - **Lip:** Contains features specific to forged videos. A near-zero FPR shows the model captures LipSync synthesis features well. However, lip features alone don't cover all lipsync dynamics (e.g., rorrelation between lip movements and head poses), and video cropping blurriness can also increase the FNR.
>
> These three branches significantly impact the model's performance. Experiments show retaining more branches increases accuracy. The Region Awareness dynamically weights three regions, adjusting the model's attention to effectively utilize features from real and fake samples, achieving a balanced performance.
>
> ---
>
> **Q1:** Is there statistics how often the latency below 100ms happen?
>
> **R2:**
> Thanks for the insightful question. According to statistical data from Akamai and Ookla, average global internet latency for broadband connections often ranges from 30 ms to 60 ms [1]. Practical observations in video conferencing support this:
>
> - Microsoft Teams: Reports network latency around 50 ms to 100 ms in optimal conditions. [2]
> - Zoom: Reports network latency often around 100 ms, with a recommended latency of 150 ms or less to avoid noticeable delays between video and audio. [3]
>
> [1] Ookla. "Speedtest Global Index – Internet Speed around the world – Speedtest Global Index." Speedtest. 2024.
>
> [2] Microsoft. "Media Quality and Network Connectivity Performance in Microsoft Teams." Microsoft. 2021.
>
> [3] Zoom Support. "Accessing meeting and phone statistics." 2024.
>
> ---
>
> **Q2:** Does this work utilize audio data or not?
>
> **R3:** Thank you for the insightful question! We followed Yang et al. [1] for data preprocessing to encode both image and audio spectrum. Here's an outline of our process:
>
> - Convert audio to spectrum using librosa.
> - Convert the power spectrum (amplitude squared) to dB units.
> - Calculate the magnification ratio $r$ for spectral alignment based on the number of video frames $N$, pixel length per frame $W$, and spectrum length $L$. Formula: $r=\frac{N\cdot W}{L}$.
> - Resize the spectrum and align 5 frames with the corresponding audio spectrum.
>
> As a result, the input samples ($I$ in Eq. (1) and Image in Fig. 4 bottom-left) contain both video and audio information. We will modify Eq. (1) to make it clearer in our revision.
>
> [1] W. Yang et al., "AVoiD-DF: Audio-Visual Joint Learning for Detecting Deepfake." TIFS. 2023
>
> ---
>
> **Q3:** Where is j index in Eq. (5)? What is none-cropped region?
>
> **R4:** Thanks for pointing it out. Here is the revised formula:
>
> \begin{equation}L_{RA}(\theta_{GR},\theta_{RA})=\sum_{j=1}^N\sum_{i=1}^{T}\frac{k}{\exp([\omega_j^i]_{max}-[\omega_j^i]_h)}\end{equation}
>
> Index $i$ denotes the frame number in one frame, and index $j$ indicates the sample currently being processed in the batch. We accumulate the losses of the $N$ samples in a batch from $1$ to $N$.
>
> None-cropped region is equal to "Head" region. (i.e., images in Fig 4. head series.)
>
> ---
>
> **Q4:** Line 185: notice from where?
>
> **R5:** Our model is primarily designed for detecting LipSync forgery. Since LipSync precisely synthesizes the shape and movement of a person's lips based on a given audio, the alterations are primarily focused on the lower part of the face [1, 2]. Therefore, we artificially constrain the model's focus, directing it to concentrate more on the regions where LipSync makes high-frequency modifications, namely the face and lip regions.
>
> [1] Guan et al. "Stylesync: High-fidelity generalized and personalized lip sync in style-based generator." CVPR. 2023.
>
> [2] Ki et al. "StyleLipSync: Style-based personalized lip-sync video generation." CVPR. 2023.
>
> ---
>
> **Q5:** Figure 8: why lip is not as important in real data?
>
> **R6:**
> As mentioned in response to your W1, both the Head and Face regions contain features necessary for determining real videos. For real samples, the lip region remains important, but its weight decreases due to the increased importance of the facial and head regions. The Region Awareness module assigns higher weights to the Head and Face regions to preserve features specific to real videos. Lip region features are maintained at a similar weight to the Face region to retain traces of forgery.
>
> ---
>
> **L1:** I can't find the limitations section.
>
> **R1:** Thank you for pointing this out! We apologize for our formatting oversight. We have discussed our limitations in Appendix E. Here are more details:
>
> - LipFD performs slightly worse on Chinese samples compared to English, likely due to pronunciation differences and training on an English dataset. We plan to include more Chinese data in future work.
> - The model underperformed on FF++ and DFDC due to being trained solely on the LipSync dataset. We plan to add DeepFake data to improve general detection capabilities.

---

> ### Comment · Reviewer_Ze8K · 2024-08-08
>
> Thank you authors for the clear rebuttal. I am increasing my rating to weak accept and I hope the paper is modified based on the rebuttal for more clarity to the readers.

---

> ### Author Response · Authors · 2024-08-08
> **Thank You for Your Positive Feedback!**
>
> Thank you so much for your positive feedback! We will ensure all your suggested modifications and additional experiments are properly included in our revision. Thank you again for the precious time and positive feedback. It encourages us a lot!

---

### Official Review · Reviewer_9Sa1 · 2024-07-03

**Soundness:** 3
**Presentation:** 2
**Contribution:** 3
**Rating:** 6
**Confidence:** 3

**Summary:**

This paper focuses on a new setting in Deepfake detection called lip-syncing fraud, which only contains fewer minor cues on the leap region. To tackle this issue, the authors provide a novel method called LipFD to obtain the features from both a global view and a regional view. Also, with the new AVLips dataset, this method shows a SOTA result compared to the recent methods.

**Strengths:**

1. This work provides a new setting on Deepfake called AVLips with a large number of high-quality samples.
2. The method mainly focuses on generating the features from the lips region which is novel.

**Weaknesses:**

Although the proposed method shows a good result, there are some confused expresses which may bring a hard understanding to readers:
1. For equation 3, what is $RA(\cdot)$ mean? What is $[F_G|\{F_R\}^i_j]$ means? There lack an explanation of these operations.
2. It will be better to have an ablation study on the selection of a vision transformer. Including the pretrain, the structure, etc.
3. It could be better to have more details about the dataset, including the number of samples, the visualization of samples with different methods, etc.

**Questions:**

See weakness.

**Limitations:**

No, the author did not include. The authors should address limitations.

---

> ### Author Rebuttal · Authors · 2024-08-07
>
> Dear Reviewer 9Sa1, we sincerely thank you for your valuable time and feedback. We are encouraged by your positive comments on our novel explorations, insightgul investigations, extensive experiments, and good motivation. We sincerely hope our following clarifications and new experiments can address your concerns.
>
> ---
>
> **W1:** For equation 3, what is $RA(\cdot)$ mean? What is $[F_G| \\{ F_R \\} {_j^i}]$ means? There lack an explanation of these operations.
>
> **R1:** Thank you for your valuable comments. $RA(\cdot)$ is our Region Awareness Module (line 183) that takes $[F_G| \\{ F_R \\} {_j^i}]$ as input and computes weights based on their content (lines 178-183). $[F_G| \\{ F_R \\} {_j^i}]$ denotes the global feature ($F_G$, line 169) concatenated with three series of region features ($F_R$, line 175) to represent the relationship between temporal features and region visual features (Fig. 4b). We will add more details in our revision.
>
> ---
>
>
> **W2:** It will be better to have an ablation study on the selection of a vision transformer. Including the pretrain, the structure, etc.
>
> **R2:** Thank you for your valuable comments. Following your helpful suggestions, we conducted an experiment regarding different pertrained datasets and structures.
>
> - With the same architecture, the parameter count has a relatively small impact on final performance. Larger pretrained datasets and more challenging pretraining tasks lead to superior model performance and more balanced recognition capabilities (reflected in small differences in False Positive Rate and False Negative Rate). **This aligns with our statement in the paper: "To effectively carry out its task (capture temporal features), the encoder necessitates extraordinary representational capacity, which can be attained through exposure to a vast number of images" (lines 158-159).**
> - Under the same pretrained dataset, more advanced model architectures typically lead to better final performances. For example, Swin Transformer achieves better results than vanilla ViTs. This is possibly because the window-based approach employed by Swin Transformer is more suitable for capturing long-term dependencies in video data, assisting in better identification of temporal features. We will further explore more effective model achitectures in our future work.
>
> | ViTs             | ACC   | AP    | FPR  | FNR  |
> | ---------------- | ----- | ----- | ---- | ---- |
> | CLIP:ViT/L14     | 95.27 | 93.08 | 0.04 | 0.04 |
> | CLIP:ViT/B16     | 95.00 | 92.05 | 0.03 | 0.07 |
> | ImageNet:ViT/L16 | 93.28 | 91.13 | 0.09 | 0.04 |
> | ImageNet:ViT/B16 | 93.27 | 91.13 | 0.09 | 0.04 |
> | ImageNet:Swin-B  | 94.66 | 92.53 | 0.06 | 0.04 |
> | ImageNet:Swin-S  | 94.59 | 90.71 | 0.02 | 0.09 |
>
> ---
>
> **W3:** It could be better to have more details about the dataset, including the number of samples, the visualization of samples with different methods, etc.
>
> **R3:** Thank you for your valuable feedback. We have provided a detailed description and visualization of the dataset in Appendix. A. Here, we continue to supplement with additional information:
>
> - Data volume:
>     - Videos: 12,000
>     - Audios: 12,000
> - Number of samples: 600,000 (up to a fifty-fold expansion ratio)
> - Forgery methods:
>     - Dynamic: Wav2Lip, TalkLip
>     - Static: MakeItTalk
> - Perturbation methods: blockwise, contrast adjustment, saturation adjustment, Gaussian blur, pixelation, compression
> - Covered scenes:
>     - Common datasets: LRS2, FF++, DFDC
>     - Real-world scenarios: video calls, streaming media
>
> Further data visualization and statistical analyses will be included in the the revision.
>
> ---
>
> **L1:** The authors should address limitations.
>
> **R4:** Thank you for this pointing out! We are deeply sorry for our formatting oversight. We have discussed our limitations in Appendix. E. Here are some more detailed explanations:
>
> - During real-world evaluation of WeChat video calls, we found that LipFD performs slightly worse on Chinese samples compared to English. We believe this is due to pronunciation differences between Chinese and English, as well as our model being trained on a purely English dataset. In future work, we plan to incorporate more Chinese corpus for training to build a general LipSync detector.
> - Our model did not achieve optimal performance on FF++ and DFDC primarily due to being trained solely on the LipSync dataset, which exhibits a significant domain gap with DeepFake. To enhance our performance, we plan to incorporate DeepFake data into our training set to develop a more comprehensive detector.

---

> > ### Author Response · Authors · 2024-08-09
> > **Thanks to Reviewer 9Sa1**
> >
> > Dear Reviewer 9Sa1:
> >
> > Please allow us to sincerely thank you again for reviewing our paper and the valuable feedback, and in particular for recognizing the strengths of our paper in terms of new setting, novel method, new dataset, and high effectiveness.
> >
> > Please kindly let us know if our response and the new experiments have properly addressed your concerns. We are more than happy to answer any additional questions during the discussion period. Your feedback will be greatly appreciated!
> >
> > Best,
> >
> > Paper6542 Authors

---

> ### Comment · Reviewer_9Sa1 · 2024-08-12
> **Review after Rebuttal**
>
> Thanks for your nice explanation of my concerns. I think this is good work and I will keep my rate as weakly accept.

---

> > ### Author Response · Authors · 2024-08-12
> > **Thank You for Your Positive Feedback!**
> >
> > Thank you so much for your positive feedback! It encourages us a lot!

---

### Official Review · Reviewer_Qots · 2024-07-12

**Soundness:** 4
**Presentation:** 4
**Contribution:** 4
**Rating:** 6
**Confidence:** 4

**Summary:**

The proposed work introduces a pioneering method for detecting lip-syncing forgery, an often overlooked threat in current research. By leveraging discrepancies between lip movements and audio signals, a dual-headed detection architecture significantly enhances detection accuracy. This work also contributes to the first large-scale audio-visual LipSync dataset, comprising nearly one hundred thousand samples, and conducts extensive experiments that demonstrate our method's efficacy. Results show up to 94% average accuracy in LipSync detection, with robust performance in real-world scenarios.

**Strengths:**

1. this work proposes a new research problem -- lip forgery detection, which is meaningful and useful. A dataset for this research problem is also proposed.

2. The anonymous github makes this work very convincing.

3. The real-life applications shown in Fig. 6 is very impressive.

**Weaknesses:**

the proposed algoritm, LipFD does not have a strong techincal novelty in learning region and global features from the multi-modal input.

**Questions:**

N/A

**Limitations:**

This method might be under-optimized solution for the facial forgery detection.

---

> ### Author Rebuttal · Authors · 2024-08-07
>
> Dear Reviewer Qots, we sincerely appreciate your precious time and valuable comments. Your positive comments of our interesting and relevant topic, clear and simple presentation, novel ideas, convincing experimental evaluation, and impressive application are very encouraging to us. We sincerely hope our following clarifications can address your concerns.
>
> ---
>
> **W1:** the proposed algoritm, LipFD does not have a strong techincal novelty in learning region and global features from the multi-modal input.
>
> **R1:** Thanks for your valuable comments. We recognize that we are based on widely used global features and multi-modal inputs. However, our method differs in the following aspects:
>
> - First, we focus on the LipSync forgery detection problem, which is different from what previous literature have explored. While we totally agree that learning region and global features from multi-modal input is a widely used strategy for different tasks [3, 4, 5, 6], we arrived at this method in a motivated, principled way, driven by our novel observations on both global temporal inconsistencies and forgery traces in local regions in LipSync forgeries. These fundamental differences set our method apart from existing techniques, despite we finally converged to similar feature extraction strategies.
> - Regarding local features, previous methods for identifying deepfakes mainly rely on single-frame images, using a single encoder to extract detailed features [1, 2]. We innovatively introduce a novel Region Awareness module to dynamically weight different region features, allowing the model to automatically focus on the more informative parts. Experimental results demonstrate that our strategy achieves excellent results.
> - Regarding global features, previous methods mostly utilize a dual-headed architecture, using separate video and audio encoders to extract features from both modalities. However, this separation of audio and video inherently introduces alignment issues between audio and video features, making it challenging for encoders to accurately align specific video segments with audio segments. We align audio and images at the fine-grained level during the preprocessing stage for the first time and use ViT's patching technique to capture the inconsistencies between the temporal lip movements and audio spectra, achieving good results.
> - We also want to respectfully mention that, beyond LipFD, we also introduced the first large scale audio-visual LipSync dataset into the community, and offered an in-depth investigation on the unique properties of LipSync forgeries. We provide these findings for the first time, and we hope our datasets and analysis can help the community better understand the unique properties and challenges associated with LipSync forgeries. We will add more discussions on this aspect in our revision.
>
> Thank you again for the helpful comments.
>
>
> [1] Haliassos, Alexandros, et al. "Lips don't lie: A generalisable and robust approach to face forgery detection." Proceedings of the IEEE/CVF conference on computer vision and pattern recognition. 2021.
>
> [2] Haliassos, Alexandros, et al. "Leveraging real talking faces via self-supervision for robust forgery detection." Proceedings of the IEEE/CVF Conference on Computer Vision and Pattern Recognition. 2022.
>
> [3] Oorloff, Trevine, et al. "AVFF: Audio-Visual Feature Fusion for Video Deepfake Detection." Proceedings of the IEEE/CVF Conference on Computer Vision and Pattern Recognition. 2024.
>
> [4] Wang, Rui, et al. "AVT2-DWF: Improving Deepfake Detection with Audio-Visual Fusion and Dynamic Weighting Strategies." arXiv preprint arXiv:2403.14974 (2024).
>
> [5] Wang, Kai, et al. "Region attention networks for pose and occlusion robust facial expression recognition." IEEE Transactions on Image Processing 29 (2020): 4057-4069.
>
> [6] Peng, Ziqiao, et al. "Synctalk: The devil is in the synchronization for talking head synthesis." Proceedings of the IEEE/CVF Conference on Computer Vision and Pattern Recognition. 2024.

---

> > ### Author Response · Authors · 2024-08-09
> > **Thanks to Reviewer Qots**
> >
> > Dear Reviewer Qots:
> >
> > Please allow us to sincerely thank you again for reviewing our paper and the valuable feedback, and in particular for recognizing the strengths of our paper in terms of pioneering method, novel, meaningful and useful research problem, convincing results, and impressive application scenarios.
> >
> > Please kindly let us know if our response and the clarifications have properly addressed your concerns. We are more than happy to answer any additional questions during the discussion period. Your feedback will be greatly appreciated!
> >
> > Best,
> >
> > Paper6542 Authors

---

### Official Review · Reviewer_K6BS · 2024-07-12

**Soundness:** 3
**Presentation:** 3
**Contribution:** 3
**Rating:** 5
**Confidence:** 4

**Summary:**

The paper introduces a novel method, LipFD, dedicated to detecting lip-syncing forgeries by exploiting temporal inconsistencies between lip movements and audio signals. This unique approach addresses a significant gap in existing DeepFake detection methods. Experimental results demonstrate that LipFD achieves high accuracy across multiple datasets, showcasing its effectiveness and robustness.

**Strengths:**

- This paper addresses a novel problem by focusing on specific DeepFake types that are challenging to detect with current DeepFake detection algorithms but perform quite well in state-of-the-art models.
- The paper is well-written and easy to follow. Experimental results indicate the effectiveness of the proposed method.
- The proposed dataset provides a solid foundation for further research in this field.

**Weaknesses:**

- The diversity of fake videos in the training set is limited, as it only includes three methods: MakeitTalk, Wav2Lip, and TalkLip. This limitation can lead to overfitting, as the classifier may easily learn the distinct patterns of these methods. For example, Wav2Lip produces blurry lip images and shows obvious artifacts when fusing lip and facial images. To demonstrate generalizability, testing on additional state-of-the-art generation methods is encouraged.
- While the method performs well on the proposed LipSync dataset, there is some variability in performance across different datasets like FF++ and DFDC. This indicates potential limitations in generalizability across diverse datasets, possibly due to the limited variety of fake videos in the training set. A robust model should be capable of detecting both LipSync and general DeepFake videos effectively.

**Questions:**

- There is a question about the spectrogram in Figure 2. How is the spectrogram obtained? From Figure 4, it seems to be the audio spectrogram. However, the audio of the fake video is real, so why are there unexpected characters like "the darkest part of the spectrum"?
- What is the meaning of "static" and "dynamic" in Line 122?
- There is a typo: LRS2 in Table 1 should be AVLips.
- Why use MakeitTalk? MakeitTalk generates the whole face instead of only the lip region, which does not align with the definition of LipSync as outlined in this paper.

**Limitations:**

No.

---

> ### Author Rebuttal · Authors · 2024-08-07
>
> Dear Reviewer K6BS, we sincerely thank you for your valuable time and comments. We are encouraged by your positive comments on our novel task, interesting idea, good writing and high effectiveness. We sincerely hope our clarifications and new experiments can address your concerns. We are happy to answer more questions and conduct more experiments if needed.
>
> ---
>
> **W1:** The diversity of fake videos in the training set is limited which can lead to overfitting. Testing on additional SOTA methods is encouraged.
>
> **R1:** Thank you for the constructive comment! We totally understand your concerns about the diversity of fake videos in our datasets. We hope the following explanations can address your concerns:
>
> - We chose MakeitTalk, Wav2Lip, and TalkLip for evaluation because they are the most representative and influential LipSync methods, covering audio-driven single image talking face generation [1, 2], contrastive learning-based approaches [3, 4], and audio-driven video lip-syncing [5, 6]. Our method, **trained only on Wav2Lip forged data**, performed well on these methods (Table 2).
> - Regarding generalizability, we **retrained LipFD on MakeitTalk, Wav2Lip, and TalkLip**, then evaluated it on the recent SOTA LipSync method, SadTalker. The results suggest our model maintains high transferability to unseen forgery methods. We will evaluate LipFD on more SOTA methods in our revision.
>
> |AUC|AP|FPR|FNR|
> |-|-|-|-|
> |94.53|99.00|0.09|0.01|
>
> [1] Zhang, Wenxuan, et al. "Sadtalker: Learning realistic 3d motion coefficients for stylized audio-driven single image talking face animation." CVPR. 2023.
>
> [2] Wang et al. Audio2head: Audio-driven one-shot talking-head generation with natural head motion. IJCAI. 2021.
>
> [3] Zhou et al. Pose-controllable talking face generation by implicitly modularized audio-visual representation. CVPR. 2021.
>
> [4] Wang et al. "Seeing what you said: Talking face generation guided by a lip reading expert." CVPR. 2023.
>
> [5] Guan et al. "Stylesync: High-fidelity generalized and personalized lip sync in style-based generator." CVPR. 2023.
>
> [6] Peng et al. "Synctalk: The devil is in the synchronization for talking head synthesis." CVPR. 2024.
>
> ---
>
> **W2:** LipFD has some variability in performance across FF++ and DFDC.
>
> **R2:** Thanks for your insightful comment! We hope our following explanations can address your concerns:
>
> - We admit that our model does not achieve the best performance on FF++ and DFDC. This is mainly because (1) Since our method is primarily targeted towards LipSync detection, we only include videos that involve LipSync in our AVLips dataset; (2) our dataset was constructed based on LRS2, which only includes head data, while the fake videos in FF++ and DFDC typically involve the whole body as well as backgrounds. Due to these domain gaps, solely training on our AVLips may lead to suboptimal performances on general DeepFakes.
> - The response to reviewer Ze8K W1 shows that features contained in face and lip regions are both utilized by LipFD, so we believe our model can detect general DeepFakes by simply enriching the training set. To verify this, we fine-tuned LipFD using 400 DFDC samples and tested it on unseen DFDC data. The result shows that LipFD achieves better performance with only a little bit more DeepFake data.
>
> |AUC|AP|FPR|FNR|
> |-|-|-|-|
> |92.50|89.31|0.07|0.07|
>
> ---
>
> **Q1:** Fig. 2: how is the spectrum obtained? Why are there unexpected characters like "the darkest part of the spectrum"?
>
> **R3:** Thank you for the insightful question. We are deeply sorry that our submission may lead you to some misunderstandings. We hope the following clarifications can address your question:
> - The spectrogram is extracted using the librosa library. It is a standard library for audio processing that is widely utilized by previous works.
> - Please kindly note that we do not mean that the audio of synced video is fake. Typically, if the spectrum of sometime is dark, it means there is a silence or very small volume at the moment [1] (i.e., the person is not talking). However, the LipSync forgery may misallocate lip opening frames (Figure 2b, frame 3) that is contradictory to the spectrum information, leading to temporal inconsistency between audio and video. Our LipFD leverages this inconsistency as a feature to detect LipSync forgeries.
>
> [1] Ilyas, Hafsa, Ali Javed, and Khalid Mahmood Malik. "AVFakeNet: A unified end-to-end Dense Swin Transformer deep learning model for audio–visual​deepfakes detection." Applied Soft Computing 136 (2023): 110124.
>
> ---
>
> **Q2:** What is the meaning of "static" and "dynamic"?
>
> **R4:** Thank you for the insightful question. Dynamic methods refer to those that take a video as input and generate a forged video (e.g., Wav2Lip, TalkLip). Static methods, on the other hand, use a single image as input to generate a video through their models (e.g., MakeItTalk, Wav2Lip with the `--static` flag enabled).
>
> ---
>
> **Q3:** There is a typo: LRS2 in Table 1 should be AVLips.
>
> **R5:** Thanks for the helpful comments. We will modify them accordingly and conduct a careful proofreading to avoid other typos in our revision.
>
> ---
>
> **Q4:** Why use MakeitTalk?
>
> **R6:** Thanks for the insightful question! We categorize LipSync methods into static (generate from a single image) and dynamic (generate from a video). MakeitTalk, a leading static method, generates realistic lip movements, expressions, and head movements. It is defined as a talking face generation method that includes LipSync [1] and is considered a LipSync method in follow-up papers [2, 3]. Therefore, we include MakeitTalk as a LipSync method in our work.
>
> [1] Zhou et al. "Makelttalk: speaker-aware talking-head animation." TOG. 2020.
>
> [2] Zhang et al. "Sadtalker: Learning realistic 3d motion coefficients for stylized audio-driven single image talking face animation." CVPR. 2023.
>
> [3] Guan et al. "Stylesync: High-fidelity generalized and personalized lip sync in style-based generator." CVPR. 2023.

---

> > ### Author Response · Authors · 2024-08-09
> > **Thanks to Reviewer K6BS**
> >
> > Dear Reviewer K6BS:
> >
> > Please allow us to sincerely thank you again for reviewing our paper and the valuable feedback, and in particular for recognizing the strengths of our paper in terms of novel problem, novel and unique approach, great significance on filling the gap, good writing, and high effectiveness and robustness.
> >
> > Please kindly let us know if our response and the new experiments have properly addressed your concerns. We are more than happy to answer any additional questions during the discussion period. Your feedback will be greatly appreciated!
> >
> > Best,
> >
> > Paper6542 Authors

---

> > ### Comment · Reviewer_K6BS · 2024-08-11
> > **Review after rebuttal**
> >
> > Thanks for the authors' detailed feedback. After reviewing the rebuttal, I find that the authors have addressed most of my concerns. Consequently, I have decided to maintain my initial rating.

---

> > > ### Author Response · Authors · 2024-08-11
> > > **Thank You for Your Positive Feedback!**
> > >
> > > Thank you so much for your positive feedback! It encourages us a lot!

---

### Decision · Program_Chairs · 2024-09-25

**Decision:**

Accept (poster)

**Comment:**

The paper received positive-leaning reviews, ranging from "Borderline accept" to "Weak Accept". The reviewers noted the novel problem of lip forgery detection in DeepFakes, which this paper raises, the dataset proposed to provide foundations for further research, the novelty of the proposed method, and the paper's real-life applications.

The reviewers were concerned with overfitting and generalization issues due to limited diversity in the fake videos used for training (K6BS). The proposed method's novelty was questioned (Qots). The reviewers found some explanations unclear (9Sa1, Ze8K) and a lack of comprehensive ablation studies was also mentioned (9Sa1, Ze8K).

These limitations were not seen as reasons for the reviewers to recommend rejecting the paper, and I don't see them as such either.